# Therapeutic itineraries and testimonies of COVID-19 patients in Manaus, the epicenter of the pandemic in the Brazilian Amazon

**Igor Castro Tavares**[1,2]*, **Denise Maria Guerreiro da Silva**[2,3], **Wagner Ferreira Monteiro**[1,2], **Kássia Janara Veras Lima**[1,2], **Darlisom Sousa Ferreira**[2], **Wuelton Marcelo Monteiro**[1,2], **Flávia Regina Souza Ramos**[2,3]

**1** Department of Research, Tropical Medicine Foundation Dr. Heitor Vieira Dourado, Manaus, Amazonas, Brazil, **2** School of Health Sciences, Amazonas State University, Manaus, Amazonas, Brazil, **3** Nursing Graduate Program, Federal University of Santa Catarina, Florianópolis, Santa Catarina, Brazil

* enf.igortavares@gmail.com

## Abstract

### Introduction

Therapeutic itineraries (TIs) involve the process of choices and the evaluation of care/treatment experiences related to the family, popular and professional subsystems in the course of an illness. The understanding of TIs is fundamental not only in the human experiential sense, but also to subsidize transformations in health care systems. The objective of the study was to describe the therapeutic itineraries of patients with COVID-19 in the city of Manaus, which had the highest mortality rate and and was the epicenter of the disease in Brazil, while considering the severity of the health crisis in the scenario.

### Method

A qualitative descriptive study that involved in-depth interviews of 29 patients who recovered from severe COVID-19 in Manaus, Brazil.

### Results

The family and popular subsystems were integrated, and the professional subsystem was more prominent, given the conditions of initial ignorance about the disease and the serious repercussions of the epidemic. The TIs covered from the initial assessment of the health condition, available resources, information, access to health services to the worsening, hospitalization and perceived impacts of fear, scientific denial and unpreparedness and collapse of the system.

**Data availability statement:** The data cannot be shared publicly due to ethical restrictions involving potentially identifiable participant information. The data are available from the Research Ethics Committee of the Amazonas State University (Universidade do Estado do Amazonas – UEA) for researchers who meet the criteria for access to confidential data. Data requests may be sent to: comite.etica@uea.edu.br.

**Funding:** Amazonas State Research Support Foundation-FAPEAM (01.02.016301.01044/2021). http://www.fapeam.am.gov.br/- The funders had no role in study design, data collection and analysis, decision to publish, or preparation of the manuscript.

**Competing interests:** The authors have declared that no competing interests exist.

## Conclusions

The reflection regarding objective and subjective behaviors of individuals in the health-disease process should last beyond the temporality of the crisis and mark the (re)construction of a resolutive health system that considers the real problems faced by users in the search for care.

## Introduction

On December 31, 2019, the World Health Organization (WHO) was alerted to several cases of pneumonia in Wuhan city, Hubei Province, in the People's Republic of China.

It was a new strain of coronavirus that had never previously been isolated from human samples and, on February 11, 2020, it was named SARS-CoV-2, which is responsible for causing the disease COVID-19. In March 2020, the WHO characterized COVID-19 as a pandemic after the disease spread across several countries [1]. By the first quarter of 2024, more than 775 million confirmed cases and more than 7 million deaths had been reported worldwide. In the same period, in Brazil, there were 38,791,997 confirmed cases and 711,964 deaths, maintaining an incidence of 18,459.4/100,000 inhabitants and a mortality of 338.8/100,000 inhabitants [2].

In the state of Amazonas, the pandemic progressed rapidly and presented two overwhelming peaks (March 2020 and January 2021), with a high number of deaths occurring in Manaus, which became the epicenter of the disease in the country [3] presenting a mortality rate 4.5 times higher [4,5]. Manaus had 642,296 confirmed cases and 14,495 deaths, with the most affected age group being between 30 and 49 years for both sexes [3]. Manaus is the largest metropolis in the Amazon, with a population of more than 2 million inhabitants and a population density of 181.01 inhabitants/km$^2$ [5]. It has been marked historically by conditions of extreme poverty and social inequalities, which are worsened by the difficult access to health services due to its extensive territorial dimensions [3]. This situation is worse for traditional populations such as riverine populations and indigenous peoples [4]. These barriers contributed to the negative highlight in facing the pandemic in Manaus. From the first confirmed case of COVID-19 in March 2020 [4,6]. Manaus faced the largest health crisis in its history, with great international repercussions [3,6]. With an unexpected explosion of new cases, the local health system collapsed, which led to overwhelming levels of contagion and mortality in the capital [4].

Despite the lack of preparedness of health services in the city of Manaus to face the pandemic, the resilience of managers promoted dynamism and technical-assistance arrangements in the field of management and humanized care [7]. Due to the numerous labor adversities caused by the pandemic to workers and managers, many urgent adaptations were necessary to deal with the voluminous and atypical demands.

There was also the experience of intense feelings of insecurity and fear during the development of staff activities on the front line of care and management during the

different phases of the pandemic [8]. Coping with the pandemic promoted a deep reflection on the functioning of the public health system in the city of Manaus, and it showed historical signs of deterioration [4,5]. Given the limitations imposed by the overwhelming burden on health services and the systemic collapse, individuals were compelled to make decisions based on subjective perceptions, fear, and shared knowledge, revealing important behavioral dynamics in the search for care.

In the health-disease process, human beings tend to seek some type of care from the moment they perceive themselves as being sick, either from a subjective perspective or from the discourse of the diagnostic rules established by biomedicine [9,10].

However, on a global scale, all countries had obstacles in diagnosis and treatment, caused either by the fear of contracting COVID-19 during the search for care [11]; lack of knowledge about the disease; feeling of fear of being labeled as having a deadly disease [6,7,12]; procrastination in seeking medical attention when the first symptoms appeared though without signs of severity [13] and, finally, the fragility of health systems in the face of difficulties related to access to diagnostic tests and the organization of the system to meet the demand of the population in the face of the pandemic [8]. These individual and collective behaviors in the search for care reflect a broader trajectory that has been conceptualized in the literature as the therapeutic itinerary.

Therapeutic itineraries refer to the paths taken by individuals in their search for health care, involving decisions, delays, actors, and care experiences across family, popular, and professional subsystems [9]. This concept provides a relevant framework for understanding how people navigate formal and informal health systems during illness, especially in times of crisis.

The understanding of therapeutic itineraries from the onset of symptoms to the care received in the health system involves the process of choices, the actors included in the process and the perception of the effectiveness of the care received. The aim of this study was to describe the therapeutic itineraries of patients with COVID-19 in the city of Manaus, during the health crisis.

## Methods

This study follows the Consolidated Criteria for Reporting Qualitative Research (COREQ) checklist, which aims to ensure transparency and completeness in qualitative studies involving interviews and focus groups. By addressing key aspects such as study design, research team, data collection, and analysis, COREQ enhances the clarity, rigor, and reproducibility of qualitative research [14].

### Ethical aspects

The study was approved by the Ethics Committee for Research with Humans of the Universidade do Estado do Amazonas (approval number 4.085.240/2020). All participants signed a consent form after reading the study objectives and procedures.

### Study design

We conducted a qualitative descriptive study with 29 patients that recovered from severe COVID-19, in Manaus, Brazil. In-depth interviews were performed in Manaus between January and December 2021. The choice of a descriptive qualitative approach is appropriate for studies such as this one since it seeks to explore and describe a complex and contextually rich phenomenon, thus providing an in-depth understanding of the experiences, perceptions and meanings attributed by participants [15].

### Study setting

#### Study team and reflexivity

The first author (ICT), a male researcher, contacted and interviewed the participants. He is a registered nurse, with training in qualitative research, and has a master's degree in Infectious Diseases. The research team was also composed of five nurses, who have extensive experience in public health and qualitative research, and one epidemiologist. The researchers had not had prior contact with any of the research participants.

## Methodological orientation

The theoretical part of this study was guided by the health care system proposed by Kleinman [16]. According to the socioanthropological literature, the trajectory that the individual carries out in the search for health care is called the therapeutic itinerary. In this trajectory, individuals or social groups mobilize various resources ranging from popular, home-based care practices to the structures and resources of biomedicine.

At this time, there are several decisions and events that aim to treat the disease, which produce important social representations in the face of difficulties, uncertainties, doubts and anxieties that involve the health dimension for all the paths taken [10–12]. The chosen methodological orientation is based on the understanding that culture provides models for human behaviors related to the health-disease process, i.e., it assumes the character of standards, norms that delimit actions and social representations in each individual's process of getting sick [9]. These systems consider elements that belong to concepts that include the collectivity of a given location in which social groups, families and individuals are part of a whole construct of social, biological and anthropological representations as factors that can determine or influence illness [9,10].

This system is formed by three interrelated subsystems that sometimes overlap due to the intersections of the paths traveled by people in the dimensions that lead to illness or the search for health care, whether of the individual, family or community. Thus, they are generically called family, popular or professional [9,10]. The family subsystem develops in the scenario of culture or common sense, in which people, their families and close members of the community interact and where healing actions are carried out based on popular, non-professional and non-specialist knowledge of biomedicine. It is the individual's first recognition of the disease and the barriers/limitations faced due to loss of homeostasis. This subsystem consists of family, friends and neighbors, i.e., all who are part of the patient's social network and who share their daily lives at the time of convalescence. It is in this subsystem that decisions are made about entering the other subsystems and it is decided, after returning to it, whether or not to follow the recommendations made by health professionals [16]. The popular subsystem includes healing specialists, but without official regulation and with limited records of their knowledge to solve problems that need scientific investigation. They can be healers or others. These specialists are widely validated by society for their healing practices, and they are strongly linked to the family subsystem [7,9,16]. They are often called laymen and their therapies involve the use of herbs, teas, spiritual surgeries, manipulative treatments and special exercises. Their cures are called sacred when it comes to shamanism and healing rituals. Anthropological studies are more focused on shamanism and healing rituals and, more recently, there has been an increase in studies of lay practices [16]. The professional subsystem is represented by the regulated and recognized professions, whose function is to promote healing [9,10,13].

## Study population and sampling

Participants or guardians of the parcipants that were under 18 years of age were included after discharge from hospitals in the public health system, mainly or exclusively for severe COVID-19. Some family members who participated in the participants' pre- or intra-hospital care were also interviewed. The sampling was non-probabilistic and was carried out in three public reference hospitals for the care of patients with COVID-19 during the pandemic. The presentation of the project and the invitation to potential participants were made on the day of discharge while in the hospital in person, or after discharge by telephone, when they were already in a better condition to report their experiences. If the patient accepted the invitation to participate in the study, an interview was scheduled at home or in another place indicated by the patient. None of the invitations were refused. The inclusion of new participants continued until data saturation was achieved [14].

## Data collection

The data were obtained through in-depth interviews, which were guided by a script that was prepared jointly by all the researchers [Table 1]. The interview script was developed based on the concept of therapeutic itineraries, aiming to capture the participants' paths, decisions, and experiences in their search for care during the course of COVID-19.

**Table 1. Interview script used in the study.**

| Item | Question |
|---|---|
| 1 | Think about the start of your COVID-19 illness |
| 2 | What did you feel that made you seek some kind of care? |
| 3 | Had you heard of this disease before? What did you know about it? |
| 4 | What care/treatment did you seek and where? |
| 5 | Who guided or helped you in this search for care/treatment? |
| 6 | How did you feel during this period of care/treatment? |
| 7 | What could have helped you to feel better? |
| 8 | Talk about your experience with COVID-19. |

The in-depth interviews were conducted in a quiet space in the home of the interviewee. Occasionally, there was a family member accompanying the visit and, at other times, only the interviewer and interviewee. The interviewer introduced himself/herself to the participant at the beginning of the interview by giving a brief personal background, explained his/her role in the study and the study's objectives. The interviews lasted for 30 minutes on average and were recorded via an audio recording device. No interviewer-related biases were identified.

### Transcript return

Participants were not invited to review their interview transcripts. This decision was made due to time constraints and the limited availability of participants for follow-up contact after the data collection phase. Moreover, the research protocol approved by the institutional ethics committee did not include transcript return as part of the study design. We considered that, given the methodological scope of the study, a rigorous and trustworthy thematic analysis could be conducted without requiring transcript revalidation by participants.

### Interviewer characteristics and reflexivity

The interviews were conducted by a doctoral student in Public Health, who is also a registered nurse, university professor, and healthcare professional affiliated with the Fundação de Hematologia do Estado do Amazonas. The researcher has 12 years of professional experience in the Brazilian Unified Health System (SUS), which contributed to contextual familiarity but did not involve clinical practice during the COVID-19 pandemic. The interviewer participated in data collection, organization, and analysis. No personal experiences with COVID-19 were reported, and no prior beliefs or biases were identified that could have influenced participant interaction or data interpretation. Reflexivity was ensured through regular discussions within the research team to support analytical neutrality and methodological rigor.

### Data analysis

The transcripts of the interviews and the field notes were imported into the Atlas.ti 8.0 software (*Qualitative Research and Solutions*) for thematic analysis based on the methodological orientation adopted. The lead researcher (FRSR) independently carried out inductive coding of the transcripts to identify and compile a preliminary list of codes. To ensure the reliability of the process, a team of four researchers analyzed and discussed these codes to arrive at a definitive set that served as a framework for analysis, the interpretation of preliminary results and the identification of key patterns. Using these insights, a second round of coding was carried out with the aim of identifying patterns in the initial coding and selecting representative quotes for each theme. These themes were then grouped into three broad categories that gave rise to the workload thematic network.

**Techniques to enhance trustworthiness**

To ensure the trustworthiness of the findings, we adopted several strategies during and after data analysis. Credibility was enhanced through analyst triangulation, with four researchers independently coding and discussing the data until consensus was reached. Dependability and confirmability were supported by maintaining an audit trail of coding decisions, ensuring transparency and consistency throughout the analytical process. Transferability was addressed by providing detailed contextual information about the participants, data collection procedures, and the study setting. These procedures aimed to strengthen the rigor of the analysis according to the trustworthiness criteria proposed by Lincoln and Guba [14].

## Results

### Characteristics of the participants

The study included 29 participants: 17 (59%) female and 12 (41%) male. Ages ranged from 4 to 67 years, with 40 years being the most frequently reported age. Regarding comorbidities, only eight participants mentioned having pre-existing conditions, with diabetes mellitus (DM) being the most common, followed by systemic arterial hypertension (SAH).

It is worth noting that 20.7% of participants were health professionals, which can be attributed to the researchers' greater access to this group as well as the high number of severe cases among them. All participants were admitted to an intensive care unit (ICU). Of the 29 interviewees, 23 did not have private health insurance and therefore received care through the public health system, while only six participants received care in private healthcare facilities.

The length of hospitalization ranged from 7 to 55 days. Among the total, seven participants required intubation, and 22 did not. The full sociodemographic and clinical profile of all participants is presented in Table 2.

The participants circulated in the three subsystems after an initial assessment of their health condition, the resources available in terms of information and the possibility of access to formal and informal health services and personal and/or family knowledge. In this search, the professional subsystem was more prominent, given the great dissemination of the pandemic situation that everyone was facing and the difficulty of finding family or popular cures for a yet unknown disease.

### Thematic synthesis of findings

Following the presentation of participant characteristics, the results are now organized into three overarching analytical themes, which reflect the core dimensions of the participants' therapeutic itineraries:

1) Initial responses to illness: family strategies, popular remedies, and shared uncertainty;

2) The paradox of the pandemic in the professional subsystem: fear, unpreparedness, and scientific denialism;

3) Emotional experiences, spirituality, and perceptions of the COVID-19 vaccine.

**Initial responses to illness: Family strategies, popular remedies, and shared uncertainty.** The first responses to COVID-19 symptoms occurred primarily within the household context. Fever, body aches, cough, and shortness of breath were often interpreted as common flu symptoms. In this setting, care decisions were made collectively by family members, who assumed an active role in monitoring the illness and managing the initial phase of care.

Without immediate access to reliable guidance and faced with growing fear, many participants turned to self-medication and popular health practices. Over-the-counter drugs and treatments promoted informally, such as ivermectin and azithromycin, were used alongside teas, medicinal plants, and traditional home remedies common in the Amazon region.

*"My husband, the one who stayed by my side all the time. And my children outside." (P12)*

*"As soon as I found out, I did what we do at home, garlic tea with lemon and lots of fluids." (P6)*

**Table 2. Profile of the study participants.**

| Patient | Age | Sex | Health insurance | Origin | Comorbidities | Number of days of hospitalization (days) | Place of hospitalization | Intubation | Income (minimum wages) | Occupation |
|---|---|---|---|---|---|---|---|---|---|---|
| P1 | 58 | M | Yes | Manaus | Hypertension and diabetes mellitus | 8 | Private | No | 1 | Retired |
| P2 | 45 | F | Yes | Manaus | Thrombophilia | 45 | Private | Yes | 8 | Businesswoman |
| P3 | 39 | F | Yes | Manaus | No | 14 | Private | No | 5 | Nurse |
| P4 | 44 | F | No | Manaus | No | 11 | Public | No | 2 | Receptionist |
| P5 | 36 | F | No | Manaus | No | 29 | Public | No | 3 | Nursing technician |
| P6 | 41 | F | No | Manaus | No | 13 | Public | No | 1 | Nursing technician |
| P7 | 68 | M | No | Manaus | DM | 10 | Public | No | 1 | Retired |
| P8 | 51 | F | No | Manaus | No | 10 | Public | No | 2 | Uber driver |
| P9 | 49 | M | No | Manaus | No | 55 | Public | Yes | 2 | Self-employed |
| P10 | 64 | M | No | Manaus | Hypertension and diabetes mellitus | 8 | Public | No | 3 | Retired |
| P11 | 61 | F | No | Manaus | No | 22 | Public | Yes | 1 | Retired |
| P12 | 51 | F | No | Manaus | No | 19 | Public | No | 0 | Housewife |
| P13 | 49 | M | No | Manaus | No | 9 | Public | No | 2 | Doorman |
| P14 | 60 | F | No | Manaus | No | 11 | Public | No | 1 | Retired |
| P15 | 48 | F | No | Manaus | No | 10 | Public | No | 5 | Teacher |
| P16 | 67 | M | No | Manaus | Hypertension and diabetes mellitus | 9 | Public | No | 1 | Retired |
| P17 | 40 | F | No | Manaus | No | 23 | Public | Yes | 1 | Self-employed |
| P18 | 41 | M | Yes | Manaus | No | 12 | Private | No | 22 | Doctos |
| P19 | 44 | F | No | Manaus | No | 9 | Public | No | 2 | Nursing technician |
| P20 | 34 | M | Yes | Manaus | No | 8 | Private | No | 7 | Nurse |
| P21 | 40 | F | Yes | Manaus | No | 16 | Private | No | 6 | Biologist |
| P22 | 4 | F | No | Manaus | Cerebral palsy | 26 | Public | Yes | 1 | Self-employed |
| P23 | 35 | F | No | Manaus | No | 12 | Public | Yes | 2 | Administrative assistant |
| P24 | 69 | F | No | Manaus | Hypertension and diabetes mellitus | 28 | Public | Yes | 1 | Retired |
| P25 | 60 | F | No | Manaus | No | 7 | Public | No | 1 | Self-employed |
| P26 | 52 | M | No | Manaus | No | 10 | Public | No | 2 | Driver |
| P27 | 64 | M | No | Manaus | DM | 8 | Public | No | 2 | Retired |
| P28 | 48 | M | No | Manaus | No | 11 | Public | No | 1 | Self-employed |
| P29 | 67 | M | No | Manaus | No | 55 | Public | Yes | 1 | Retired |

These actions reflect the coexistence of biomedical and traditional knowledge within everyday care practices. Although lacking scientific validation, such strategies provided a sense of agency and emotional reassurance in the face of a poorly understood and rapidly evolving disease.

*"Additional participant quotes related to this theme are provided in the supplementary file."*

**The paradox of the pandemic in the professional subsystem: Fear, unpreparedness, and scientific denialism.** The transition to the professional health subsystem often occurred after the failure of initial care at home.

However, participants encountered fragmented, delayed, or inconsistent access to medical services, reflecting widespread disorganization and resource scarcity.

Several participants faced barriers to diagnosis due to limited testing or clinical misinterpretation of symptoms. These gaps frequently contributed to late-stage admissions and worsened clinical outcomes.

*"The test wasn't available, I only managed to get tested because I knew someone working in healthcare." (P5)*

Once within the system, participants described contradictory protocols and unvalidated treatments, including the prescription of hydroxychloroquine and ivermectin, reflecting the influence of politicized discourses on care delivery.

*"They gave me that standard protocol: COVID, chloroquine, azithromycin… and I took it." (P22)*

Hospitalization was often described as traumatic. During peak waves, overcrowding, lack of beds, and the collapse of oxygen supply systems created fear and a sense of helplessness.

*"We couldn't get it. It was the night when all the ICU patients died without oxygen." (P1)*

These experiences revealed systemic unpreparedness and denialism within institutional responses, contributing to mistrust in public health systems and deepening suffering.

*Additional participant quotes related to this theme are provided in the supplementary file.*

**Emotional experiences, spirituality, and perceptions of the COVID-19 vaccine.** The psychological impact of COVID-19 was a central theme in participants' narratives. Fear of death, the trauma of witnessing others die, and the loneliness of prolonged hospitalization were recurring elements, particularly among those who stayed in intensive care.

*"What really affects us there is the psychological aspect; there were people dying beside us." (P1)*

Feelings of isolation were intensified by hospital visitation bans and uncertainty about prognosis. In response, many turned to prayer or reflected on existential questions, especially after surviving severe illness.

*"I think it was so many prayer groups... sometimes I ask: God, why did I survive and so many others died?" (P2)*

Participants' views on the COVID-19 vaccine were influenced by their illness experiences but also shaped by religious and political discourse. While many embraced vaccination, others initially hesitated due to misinformation, particularly from religious or political leaders.

*"I'm evangelical, but I didn't let that interfere... I heard pastors saying it was the mark [of the beast]." (P3)*

The findings underscore how decisions regarding vaccination were not purely biomedical but intersected with broader moral, ideological, and spiritual considerations, highlighting the need for inclusive and culturally sensitive health communication strategies.

*Additional participant quotes related to this theme are provided in the supplementary file.*

## Discussion

The analysis of the TIs in search of health care during the COVID-19 pandemic is not simply the difficulties of access to the health system, but it can reveal the path taken, the decisions made in this search for care and the evaluations made about such decisions. The mutual relations between the family, popular and professional subsystems were evidenced and produced psychological and social repercussions that extrapolate the moment of the disease itself. In this sense, it was decided to discuss the categories from diverse elements linked to each subsystem.

### The COVID-19 therapeutic itinerary of the population of Manaus

The choices that marked the TIs could not be discussed outside the confluence of complex social, personal and family factors. The concrete conditions and the crisis of the health system, defined within the professional subsystem, test the limits and make the boundaries between the subsystems more fragile and clearer. This may be a differential result of the TI study in the specific case of COVID-19. Precariousness and insecurity are also shown in the path and in the decisions of each person. In addition, reports from users and professionals converge in a journey of fear – experiences transformed by knowledge, by the growing proximity to death/loss and with their own personal transformations in the face of the threat [17].

The therapeutic itinerary addresses a succession of events of the illness process and is not separated from the aspects of a person's daily life. Therefore, it is in the family subsystem where the sick person and their family first perceive and experience the symptoms, label and evaluate the disease and engage in a specific type of behavior in the search for a cure [16]. This path takes place with decisions and interpretations derived from common sense, which are made individually or with the help of family members and a network of friends [8,9]. Before seeking the professional system, most people follow steps and take decisions to treat and alleviate the signs and symptoms of the clinical course of their disease.

In this system, to the inequalities that shape family life and its impossibilities [18], we can add the specific vulnerabilities to the limits of access to rights and information [19–24]. The pandemic has more vehemently exposed fragile social protection while serious restrictions on rights and qualified information have expanded to the whole of society. Hence, the interest in the analysis of the TIs in situations of health crises must remain beyond the temporality of a pandemic, to recognize both the exceptionality experienced by all and the differences in the path of each one.

The experience of the disease is dynamically configured in the social space – the choices and paths express both individual and collective constructions that are shaped by multiple influences [25]. This showed itself in the form of surprise and fear from the first physical manifestations of the disease. Being affected by a new and little-known disease generated a relative initial immobility.

Given the limiting circumstances of the pandemic, the participants valued the help of the closest people as an important resource in the decision to solve their problems. From the first symptoms, through diagnosis and hospitalization, family support was fundamental to access services and even fight for omitted resources.

The use of medicinal plants occurred, even if there was not yet a recognized arsenal of practices of the popular subsystem for COVID-19. This can probably be explained by the intense social representation that medicinal plants have had in Amazonian populations over the generations, strongly marking the construction of this society [26,27]. The use of medicinal plants is a traditional practice among the Brazilian population [26,28] and was reported by the participants as a first way to mitigate the manifestations of COVID-19. In addition to representing a possible "resistance to domination, demarcation and normalization of bodies in health care processes" [25], traditional practices demonstrate circulation in different health care subsystems, in a nonlinear way, one resource not excluding the other. The therapeutic itinerary, therefore, goes beyond the biological and technical dimensions of care and includes the dimension of human and symbolic relationships [25].

Circulation in the family and popular subsystems was described in Algeria, when panic over an insufficient professional system reinforced forms of support and solidarity between families in the use of traditional resources [29].

Uncertainties about treatment also made people resort to self-medication, which saw a considerable increase during the pandemic. In addition to the fear of getting sick, the fact that they are included in the treatment protocol of the Ministry of Health contributed to the quadrupled sales of some drugs, such as ivermectin and vitamin D [16].

Self-medication intensified due to misinformation, treatment uncertainties and ease in acquiring drugs such as chloroquine and hydroxychloroquine, which have no scientifically proven efficacy or safety, even as a preventive measure [30,31].

The internet and social networks proved to be an important source of information and a communication channel so that updates regarding COVID-19 could reach the lay citizen, despite different conditions of access to virtual tools. The phenomenon of an infodemic [32,33] or "information epidemic" leveraged the dissemination of different data, interpretations and speculations, which increased fear, despair and (dis)information that impacted epidemiological responses to the pandemic.

Aggravating the issue of misinformation was the increasing politicization of the fight against the pandemic, since some world leaders (such as in Brazil) refused or opposed the adoption of isolation as a preventive method and still defended the use of drugs not recommended for the management of the disease in an attempt to minimize the severity and find a quick and cheap solution [34]. In addition, the spread of denialist behaviors was observed, against science and against the recommendations made by the WHO or qualified information, with the sharing of false news on social networks [35].

During the COVID-19 pandemic, the pattern of drug consumption in Brazil drew attention. The prescription of drugs to treat or prevent COVID-19 without scientific evidence received contours of great credibility, when "early treatment" and "the COVID kit" were disseminated and their use encouraged widely on social media, including by medical professionals, public authorities and on the official internet pages of Health Secretariats, the Ministry of Health and the Federal Government of Brazil [36,37]. This may help explain the fact that this type of treatment is not limited to the family subsystem but is maintained and continued in the professional subsystem. In fact, this is another differential of the TIs of the COVID-19 pandemic, when compared with that of other diseases, which highlights the importance of this type of study and its applicability in guiding system preparation strategies to prevent avoidable deaths in future emergencies.

In this sense, it is essential that the authorities responsible for conducting health systems use the experiences they had in the COVID-19 pandemic as a starting point for more assertive, safe and well-informed decisions and forms of treatment and prevention.

The first actions of the TI have repercussions not only in an isolated family context but in a whole social and economic ecosystem that needs to be analyzed and prepared. The course of clinical treatment and care in the face of diseases has a strong relationship with the capacity for discernment and the quality of information. In order for the scientific bases to be incorporated into care practices, treatments and preventive attitudes, language and forms of interlocution must certainly approach the universe of meanings and experiences of personal, family and community life in the various social groups.

The professional subsystem is formed by the systems, services and health professions, which are legally validated and recognized in the structures of cures in the biomedical dimension, as a source of hegemonic power in society, which influences the decision-making of individuals [37].

This subsystem was the subject of choices and evaluations of the study participants in the search for health care. The time of worsening of the clinical condition was decisive for the entry into this subsystem, either in public or private health services, usually starting from the family subsystem, i.e., by a decision made with the help of the family [38].

This search was like that found in other studies on the therapeutic itinerary of people with COVID-19, whose initial movement was the search for diagnosis, based on the clinical picture or imaging tests [8]. However, the results differed regarding the fact that the diagnostic confirmation marked the beginning of the therapeutic itinerary. In the case studied, the definitive diagnosis happened at different times of the itinerary, sometimes in situations of severe clinical condition, also due to the lack of access to laboratory tests.

It is worth noting that the TIs of the participants revealed that even after moving to the family or popular subsystem, the professional system was the alternative in the face of worsening of the clinical picture. This trajectory was not always linear but was marked by comings and goings – from home to the services – either by impossibility of care due to exhaustion

of health services, by the collapse of the system or by the difficulty in diagnosis, even with persistent and worsening symptoms [P5, P8, P17, P28].

Another negative point was the exacerbation of the unpreparedness of the health system in the face of great demand for care. This condition was made clear by the impossibility of attending the population, lack of supplies, clinical beds and intensive care unit, health professionals and especially the lack of oxygen in the public hospital network of Manaus, which led to death by asphyxiation in dozens of patients who depended on oxygen for life support [39]. The facts that shocked the Brazilian population and the world had repercussions in the interior of the state of Amazonas due to the difficulties in access and precarious hospital medical infrastructure.

Inevitably, the network of events related to the unpreparedness of the health service generated an intense wave of fear of death by the general population and especially by people with a positive diagnosis for COVID-19. Regardless of the severity of the clinical picture, the feeling of fear and helplessness was inevitable and constant.

This sentiment that was anchored in alarming data, especially from December 2020 to January 2021, the exact periods that the public and private health systems of Manaus collapsed. In the period from April to December 2020, 3,380 deaths from COVID-19 were notified in the city of Manaus, while only in January 2021 2,195 deaths were notified, evidencing a new and avoidable peak in mortality [39].

Amid all the chaos and uncertainties experienced by the population, the emergence of a vaccine that would combat the disease represented the hope of a future that was free of COVID-19. However, there was a great wave of misinformation and denial of science, propagated by various information vehicles and social networks [35].

Although most of the Brazilian population believes in the effectiveness of the vaccine, it is worrying how the (dis)information conveyed is powerful and easily disseminated throughout society. There is a minority, albeit an expressive one, that positions itself as "antivaccine". This movement is worrying from a health point of view, showing that within the TIs the decision on whether to vaccinate or not has to be a constant agenda among health authorities, so that effective communications are produced and reach everyone.

It is up to health management bodies to face this challenge, recognizing the different ideological and historical elements present in movements with any association with negationism. In a different historical moment, the vaccine revolt (Rio de Janeiro, 1904), for example, refers to motivations for violence, authoritarianism and arbitrariness in the fight against diseases, while the denial of the COVID-19 pandemic had updated political interests [40]. The traces of aggressiveness and political polarity of current denialism, whether Brazilian or world denialism, is exacerbated by misinformation. In the Brazilian case, it differs from the vaccine revolt in terms of the position of disdain or encouragement by the federal government/bolsonarism itself, while the position opposed to the government by the mainstream media had different meanings in both moments – support for the vaccine (COVID-19) or rejection of vaccination/obligation (1904). In the case of COVID-19, the obligation was not in fact instituted but was only used to create a sense of threat to the loss of freedom. The current movement is marked by the discourse based on mobilization through fear/discredit to immunizers, distrust of public health institutions with false moralism in favor of family power (offense to the honor of the head of the family) and the vaccine as a cause of death in opposition to a false and conservative defense of freedom [41].

While demonstrations suffocated the process of scientific and health security of a nation, unfounded statements denied the funeral collapse in Manaus and attacked Amazonian scientists who researched the real effects of chloroquine in the treatment of COVID-19.

## The health system in Manaus: Before and after the pandemic

The escalation of the pandemic generated a great impact, both direct and indirect, on health services around the world due to the sudden and unexpected increase in medical care, hospitalizations with high complexity assistance requirements and the overload of public and private services in terms of inputs, equipment and especially health professionals [42].

In Brazil, health is legally assumed as a right of everyone and a duty of the state, guaranteed through social and economic policies aimed at reducing the risk of disease and other problems and universal and equal access to actions and services for its promotion, protection and recovery [43]. Despite the scope and credibility of the Unified Health System (SUS), on which 80% of the population depends [44], the national management was decisive for the outcome in facing the worst crises in the history of the Brazilian health system.

This negative dynamic of the response to the pandemic was certainly worsened by Brazil being a country with continental-sized geographical dimensions, which also influences social inequality and historical health problems, especially regarding communicable diseases [45]. The pandemic reproduced social inequalities already present in the Brazilian context by reaching the most vulnerable population with greater impact [42].

The principle of universality of the SUS was not guaranteed to all Brazilian citizens, given the high lethality throughout Brazil and especially in the northern and northeastern regions of the country.

The COVID-19 pandemic put unprecedented pressure on several dimensions of the Brazilian health system, exacerbating existing problems, such as the number of hospitals equipped with intensive care units (ICU) and respirators, as well as enough trained professionals for the demand in all regions of the country [42].

The population of Amazonas is already affected by the lack of basic sanitation, ineffective primary care, lack of or poor distribution of clinical and ICU beds among municipalities, health policies that do not reach traditional populations, high transmission rates of sexually transmitted infections, and centrality of health actions in the capital, among other problems; which made the impacts of the pandemic more evident and more harmful to the entire society [46].

With the COVID-19 pandemic, the state went through one of the worst moments in the country, with a lack of medical oxygen leading to the death of 31 people at the peak of a conglomeration of problems in the health system in Manaus.

This problem suggests that the lack of or mismanagement of health services is the main cause of the resounding collapse experienced during the pandemic [47]. Despite the exposure of the problems exacerbated during the pandemic and all the popular and media pressure faced with the chaos in the health system of the state of Amazonas, improvements in the state or local health systems produced by this learning have still not been perceived.

The study points out critical elements of the professional subsystem that are not limited to the temporality of the pandemic; that only made its configuration voluminous and tragic. most inpatient and complementary beds are concentrated in Manaus, which also overloads the health services of the Amazonian capital. Despite the number of beds having increased in the state (1.47 beds/1,000 inhabitants) and in the capital (1.61 beds/1,000 inhabitants) after the pandemic, they are still below the Brazilian average of 2.1 beds/1,000 inhabitants and the WHO recommendation of 3–5 beds/1,000 inhabitants [48]. Solid improvements in the public health system are not yet evident. The crisis in the health system of the state of Amazonas in the pandemic certainly incurred on the lethality of COVID-19, which was exponentiated by limited access to health services, by political dynamics and by the higher incidence of the disease in groups with lower socioeconomic status [49], by lack of effective isolation, housing in densely occupied areas, poor sanitary conditions and a health education process not rooted in the Amazonian context [39,50,51].

Although the COVID-19 pandemic exposes the weaknesses of the health system of the state of Amazonas, which is marked by reactive and discreetly preventive health actions, the horrors experienced by the population and expressed in the individual testimonies reported in this study should drive organizational changes for greater commitment to the universal principles of the SUS, with expansion and qualification of access to health services.

## Limitations and directions for future research

While this descriptive qualitative study offers valuable insights into the therapeutic itineraries of patients with COVID-19 in Manaus, certain methodological considerations should be acknowledged. The study was conducted in a single urban setting – Manaus, the epicenter of the pandemic in Brazil – which may limit the generalizability of the findings to other regions with different healthcare infrastructures or sociocultural profiles.

 

Data were collected retrospectively through interviews conducted after participants had recovered. This approach may be subject to recall bias; however, the intensity of the experiences during the pandemic likely contributed to the vivid recollection of significant events. In addition, although the sample was diverse, it was not stratified by criteria such as age, education level, or socioeconomic status, which may restrict broader comparative analyses.

Despite these limitations, the study employed recognized strategies to ensure methodological rigor in qualitative research, including analytical triangulation, detailed contextual description, and transparency in the interpretive process. The findings contribute to a deeper understanding of the challenges faced by the population during the collapse of the health system and offer an evidence base to inform public health policies.

Future research may benefit from comparative studies involving different regions and population groups, particularly rural, Indigenous, or riverine communities that face unique structural barriers to healthcare access. Longitudinal studies may also be valuable in exploring how care-seeking behaviors evolve over time during prolonged public health crises.

Finally, we acknowledge that participant validation of the thematic findings (member checking) was not conducted. This decision was based on time constraints and the unavailability of participants for follow-up contact. Although this step was not part of the approved research protocol, we consider that the absence of participant checking did not compromise the interpretive rigor of the study. Trustworthiness was maintained through team triangulation, dual coding, consensus-based thematic analysis, audit trail documentation, and ongoing reflexivity throughout the analytic process. Moreover, participants' narratives were detailed, reflective, and voluntarily expressed, allowing for deep interpretative engagement despite the lack of formal transcript or theme validation.

## Conclusions

From the paths taken by the users in the search for cure/treatment of COVID-19, it was possible to identify the route and their experiences and representations in each health subsystem (popular, family and professional). It is imperative to report that the family and popular subsystems in this study appear as symbiotic elements in the face of COVID-19. The attitudinal actions of the first symptoms were combated with medicinal plants offered by or taught to them by family and friends simultaneously, and these same actors participate in the entire trajectory of the search for a cure together with the sick person, i.e., the entrance and exit from or to each subsystem.

The understanding of this parental or friendship relationship is fundamental to in order understand the objective and subjective behavior of individuals in the health-disease process, as well as social groups, whether large or small. Based on the understanding of such phenomena, it is possible to objectify health actions that promote the preventive and health care dimensions of the population; perhaps strengthening health services that are more preventive than reactive in the face of health problems.

In this sense, the analysis of the professional subsystem followed by users during the pandemic revealed several gaps in care and organizational gaps that were decisive for causing numerous problems for users. The need to create mechanisms that can manage the health demands of the city of Manaus was obvious.

The COVID-19 pandemic revealed to the world how fragile health systems are and how fallible society is in the lack of health actions based on science and human behavior. The city of Manaus was supposedly marked in world history by one of the darkest events of the pandemic, which was the death of dozens of people asphyxiated by a lack of oxygen. The researchers of this study believe that these negative memories are pillars for a (re)construction of a consolidated health system, decentralized and mainly focused on the real needs of the populations and that they should strongly consider the TIs of the users in search of a cure, treatment or hospital medical care for their illness.

## Supporting information

**S1 File. Supplementary participant quotes.** This file contains additional verbatim quotes from study participants, organized according to the three main analytical themes presented in the Results section: 1. Initial responses to illness: family strategies, popular remedies, and shared uncertainty; 2. The paradox of the pandemic in the professional subsystem: fear,

unpreparedness, and scientific denialism; and 3. Emotional experiences, spirituality, and perceptions of the COVID-19 vaccine.
(DOCX)

## Acknowledgments

The authors acknowledge the use of ChatGPT (OpenAI, 2024 version) as an editorial support tool during the preparation of this manuscript. The tool was used specifically to enhance linguistic clarity, textual cohesion, and structural organization. All contributions generated by artificial intelligence were carefully reviewed, validated, and approved by the authors to ensure the accuracy and scientific integrity of the work.

## Author contributions

**Conceptualization:** Denise Maria Guerreiro da Silva, Darlisom Sousa Ferreira, Wuelton Marcelo Monteiro, Flávia Regina Souza Ramos.

**Data curation:** Igor Castro Tavares, Darlisom Sousa Ferreira, Wuelton Marcelo Monteiro, Flávia Regina Souza Ramos.

**Formal analysis:** Igor Castro Tavares.

**Funding acquisition:** Darlisom Sousa Ferreira, Wuelton Marcelo Monteiro, Flávia Regina Souza Ramos.

**Investigation:** Igor Castro Tavares, Denise Maria Guerreiro da Silva, Kássia Janara Veras Lima.

**Methodology:** Igor Castro Tavares, Denise Maria Guerreiro da Silva, Darlisom Sousa Ferreira, Wuelton Marcelo Monteiro, Flávia Regina Souza Ramos.

**Project administration:** Igor Castro Tavares, Wuelton Marcelo Monteiro, Flávia Regina Souza Ramos.

**Resources:** Wuelton Marcelo Monteiro.

**Supervision:** Igor Castro Tavares, Denise Maria Guerreiro da Silva, Wagner Ferreira Monteiro, Darlisom Sousa Ferreira, Flávia Regina Souza Ramos.

**Validation:** Igor Castro Tavares, Flávia Regina Souza Ramos.

**Writing – original draft:** Igor Castro Tavares, Denise Maria Guerreiro da Silva, Wuelton Marcelo Monteiro, Flávia Regina Souza Ramos.

**Writing – review & editing:** Igor Castro Tavares, Denise Maria Guerreiro da Silva, Kássia Janara Veras Lima, Wuelton Marcelo Monteiro, Flávia Regina Souza Ramos.

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
