## [Decision Letter · Decision Letter 0]

Therapeutic Itineraries and Testimonies of COVID-19 Patients in Manaus, the Epicenter of the Pandemic in the Brazilian Amazon

PLOS ONE

Dear Dr. Tavares,

Thank you for submitting your manuscript to PLOS ONE. After careful consideration, we feel that it has merit but does not fully meet PLOS ONE’s publication criteria as it currently stands. Therefore, we invite you to submit a revised version of the manuscript that addresses the points raised during the review process.

Both referees offered suggestions to improve the manuscript. I suggest you follow them. 

We look forward to receiving your revised manuscript.

Kind regards,

Tiago Pereira

Academic Editor

PLOS ONE

Journal Requirements:

3. We note that your Data Availability Statement is currently as follows: All relevant data are within the manuscript and in Supporting Information files.

Reviewers' comments:

Reviewer's Responses to Questions

**Comments to the Author**

1. Is the manuscript technically sound, and do the data support the conclusions?

Reviewer #1: Yes

Reviewer #2: Yes

2. Has the statistical analysis been performed appropriately and rigorously?

Reviewer #1: Yes

Reviewer #2: N/A

3. Have the authors made all data underlying the findings in their manuscript fully available?

Reviewer #1: Yes

Reviewer #2: No

4. Is the manuscript presented in an intelligible fashion and written in standard English?

Reviewer #1: Yes

Reviewer #2: Yes

Reviewer #1: Thank you for the opportunity to review this paper on the therapeutic itineraries of COVID-19 patients and thanks to the authors for the effort they put into this work. Below are comments and recommendations that may help enhance the paper further.

Minor issue

No page numbering in the paper, which makes it difficult to map comments with pages. kindly include page numbering

Major issues

Introduction: The introduction needs improved flow between paragraphs. For instance, the transition from paragraph 3 to paragraph 4 does not effectively connect the discussion of health systems/services to human health-seeking behaviour. Including a transition sentence that bridges these topics would improve clarity. Similarly, paragraphs 4 and 5 need a clear link to establish their relationship. Currently, these paragraphs appear stand alone. Furthermore, it is important to clearly define "therapeutic itinerary" before discussing its implications to strengthen the overall narrative, probably described early in the introduction.

Method: It is obvious that the report of this study adheres to a reporting guideline, though it is not documented. At the beginning of this section, please indicate which reporting guideline the study follows.

Could it be that you missed the information under the study setting or heading levels issue? Readers should be able to easily distinguish between different levels of headings. For example, the "study setting" and its subsections are currently using the same heading level.

Data collection: Clearly add a statement that the script for interviews was developed in line with TI.

You may want to include a subsection around techniques used to enhance trustworthiness after data analysis.

Results: The findings are interesting, but at present, they appear to lack sufficient synthesis, with brief statements and numerous quotes. Please synthesis the results under the three main themes and/or relevant subthemes. For each theme or subtheme, one or two carefully selected quotes should suffice following a thorough synthesis of the findings. If there are additional noteworthy quotes you'd like to include, consider adding them as a supplementary file.

After presenting participants' characteristics, please begin a new subsection that marks the start of the thematic presentation and synthesis of findings theme by theme.

Discussion: I found this section too long especially the “The COVID-19 therapeutic itinerary of the population of Manaus” subsection, though very rich content. I was wondering if the discussion section could be reduced to make it more concise.

Include a clear subsection on limitations of the study and potential for future research.

Best wishes.

Reviewer #2: Dear Authors,

Thank you for submitting your manuscript. Your qualitative exploration of patients’ experiences during the pandemic offers meaningful contributions to understanding healthcare navigation in crisis contexts. While the study aligns broadly with the journal’s requirements, revisions are necessary to ensure methodological transparency and compliance with ethical and reporting standards. Below, I outline specific feedback to strengthen your work.

1. Data Availability Statement Clarification

Your current statement asserts that "Yes - all data are fully available without restriction", but this may not align with ethical obligations for sensitive qualitative data. Given the sensitive nature of interview transcripts and audio recordings, public deposition could risk participant confidentiality. To resolve this, please revise the Data Availability Statement to explicitly acknowledge these ethical restrictions. This adjustment complies with PLOS ONE’s data policy, which permits restricted access for ethically sensitive datasets.

2. Adherence to Qualitative Reporting Guidelines (COREQ/SRQR)

The manuscript generally follows PLOS ONE’s standards for qualitative research but requires additional details to meet the COREQ (Consolidated Criteria for Reporting Qualitative Research) or SRQR (Standards for Reporting Qualitative Research) frameworks. Key gaps include:

Interviewer Characteristics and Reflexivity (COREQ Item 8): A discussion of the interviewers/research team background, assumptions, and potential biases is absent. For instance, were interviewers clinicians, social scientists, or individuals with personal ties to COVID-19 care? Detailing their professional roles, motivations for conducting the study, and any preconceptions (e.g., beliefs about healthcare access) would contextualize how their perspectives may have influenced participant interactions, data interpretation, or thematic analysis.

Transcript Return (COREQ Item 23): The manuscript does not mention whether participants were offered the opportunity to review their interview transcripts. This practice, common in qualitative research, enhances accuracy and participant agency. If transcript return was not feasible (e.g., due to time constraints or participant availability), briefly justify this decision in the Methods section.

Participant Checking (COREQ Item 28): There is no description of whether findings were validated with participants (member checking). If this step was omitted, acknowledge it as a limitation and discuss how its absence might affect the interpretation of results (e.g., potential discrepancies between researcher and participant perspectives).

3. Final Recommendations

Addressing these points will bolster the study’s rigor and transparency. I encourage you to consult the COREQ checklist or SRQR guidelines to ensure comprehensive reporting. Minor revisions to the Methods and Discussion sections—particularly clarifying researcher reflexivity and ethical protocols—should suffice.

Thank you for your dedication to this important topic. We look forward to reevaluating your manuscript upon resubmission.

Best regards

**Do you want your identity to be public for this peer review?** For information about this choice, including consent withdrawal, please see our Privacy Policy

Reviewer #1: **Yes: ** Oluwatoyin Adeniji

Reviewer #2: No

---

## [Author Response · Author response to Decision Letter 1]

21 May 2025

We thank the reviewers and the editor for their constructive feedback. All comments were addressed carefully, and corresponding changes were made to the manuscript. A detailed point-by-point response is provided in the file “Response to Reviewers”.

---

## [Decision Letter · Decision Letter 1]

Itinerários Terapêuticos e Depoimentos de Pacientes com COVID-19 em Manaus, Epicentro da Pandemia na Amazônia Brasileira

PONE-D-24-53418R1

Dear Dr. Tavares,

We’re pleased to inform you that your manuscript has been judged scientifically suitable for publication and will be formally accepted for publication once it meets all outstanding technical requirements.

Kind regards,

Tiago Pereira

Academic Editor

PLOS ONE

Additional Editor Comments (optional):

Reviewers' comments:

Reviewer's Responses to Questions

**Comments to the Author**

Reviewer #1: All comments have been addressed

2. Is the manuscript technically sound, and do the data support the conclusions?

Reviewer #1: Yes

3. Has the statistical analysis been performed appropriately and rigorously?

Reviewer #1: Yes

4. Have the authors made all data underlying the findings in their manuscript fully available?

Reviewer #1: Yes

5. Is the manuscript presented in an intelligible fashion and written in standard English?

Reviewer #1: Yes

Reviewer #1: Thank you for addressing the comments. Few observations below:

Abstract: Check typo error “which had the highest mortality rate and and was the epicenter of the disease in Brazil”.

Study team and reflexivity: “The first author (ICT), a male researcher” I'm not sure it's necessary to include the researcher's gender unless it has a clear implication for the interviews. If it does, then it would be important to explain how the researcher's gender may have influenced the data collection or interpretation.

Please merge the "Study Team Characteristics" and "Interviewers’ Characteristics" sections under a single section; perhaps titled "Researchers’ Characteristics" and position it where the "Study Team Characteristics" section is currently located.

Transcript return: “This decision was made due to time constraints”: What is the reason for time constrain? I could see the study is part of a PhD, is the time constraint due to PhD timeline? If so, kindly state this.

I'm unsure where the supplementary file containing the additional quotes was referenced. I believe it would be more appropriate to reference it under the "Thematic Synthesis of Findings" section.

Good luck.

**Do you want your identity to be public for this peer review?** For information about this choice, including consent withdrawal, please see our Privacy Policy

Reviewer #1: **Yes: ** Adeniji

---

## [Editor Report · Acceptance letter]

PONE-D-24-53418R1

PLOS ONE

Dear Dr. Tavares,

I'm pleased to inform you that your manuscript has been deemed suitable for publication in PLOS ONE. Congratulations! Your manuscript is now being handed over to our production team.

Kind regards,

on behalf of

Dr. Tiago Pereira

Academic Editor

PLOS ONE